# Reviewing the Review: A Pilot Study of the Ethical Review Process of Animal Research in Sweden

**DOI:** 10.3390/ani11030708

**Published:** 2021-03-05

**Authors:** Svea Jörgensen, Johan Lindsjö, Elin M. Weber, Helena Röcklinsberg

**Affiliations:** 1Department of Animal Environment and Health (HMH), Swedish University of Agricultural Sciences, P.O. Box 7068, 750 07 Uppsala, Sweden; svea.jorgensen@slu.se (S.J.); johan.lindsjo@slu.se (J.L.); 2Department of Animal Environment and Health (HMH), Swedish University of Agricultural Sciences, P.O. Box 234, 532 23 Skara, Sweden; elin.weber@slu.se

**Keywords:** ethical review, animal ethics committee (AEC), harm–benefit analysis (HBA), harm–benefit, 3R, animal research, animal welfare, animal ethics

## Abstract

**Simple Summary:**

The use of research animals is regulated within the EU through Directive 2010/63/EU on the protection of animals used for scientific purposes, as well as through national legislations and guidelines. However, the ethical review process, which all animal research must undergo, has been heavily criticized. This pilot study has analyzed the ethical review process in Sweden, focusing on how well legislative demands are fulfilled by researchers and animal ethics committees. After developing a score sheet, 18 documents (including both applications and decisions) were thoroughly reviewed, and the requests in the application form were compared to legal demands. The results revealed a number of issues concerning how HBA (harm–benefit analysis) was conducted by the committees, application, and review of the 3Rs (Replace, Reduce, Refine), as well as how humane end-points, severity assessment, and the “upper limit” of suffering were implemented and assessed. The study further indicates disconcerting discrepancies between the Swedish application forms for project evaluation, national legislation, and the directive as well as a lack of transparency throughout the review process. These findings risk compliance with the directive, animal welfare, research validity, and public trust. Therefore, a number of suggestions for improvements are provided, and the need for further research is emphasized.

**Abstract:**

The use of animals in research entails a range of societal and ethical issues, and there is widespread consensus that animals are to be kept safe from unnecessary suffering. Therefore, harm done to animals in the name of research has to be carefully regulated and undergo ethical review for approval. Since 2013, this has been enforced within the European Union through Directive 2010/63/EU on the protection of animals used for scientific purposes. However, critics argue that the directive and its implementation by member states do not properly consider all aspects of animal welfare, which risks causing unnecessary animal suffering and decreased public trust in the system. In this pilot study, the ethical review process in Sweden was investigated to determine whether or not the system is in fact flawed, and if so, what may be the underlying cause of this. Through in-depth analysis of 18 applications and decisions of ethical reviews, we found that there are recurring problems within the ethical review process in Sweden. Discrepancies between demands set by legislation and the structure of the application form lead to submitted information being incomplete by design. In turn, this prevents the Animal Ethics Committees from being able to fulfill their task of performing a harm–benefit analysis and ensuring Replacement, Reduction, and Refinement (the 3Rs). Results further showed that a significant number of applications failed to meet legal requirements regarding content. Similarly, no Animal Ethics Committee decision contained any account of evaluation of the 3Rs and a majority failed to include harm–benefit analysis as required by law. Hence, the welfare may be at risk, as well as the fulfilling of the legal requirement of only approving “necessary suffering”. We argue that the results show an unacceptably low level of compliance in the investigated applications with the legal requirement of performing both a harm–benefit analysis and applying the 3Rs within the decision-making process, and that by implication, public insight through transparency is not achieved in these cases. In order to improve the ethical review, the process needs to be restructured, and the legal demands put on both the applicants and the Animal Ethics Committees as such need to be made clear. We further propose a number of improvements, including a revision of the application form. We also encourage future research to further investigate and address issues unearthed by this pilot study.

## 1. Introduction

Animal research procedures are defined in Article 3 p. 1 of Directive 2010/63/EU (henceforth known as “the Directive”). The definition includes the creation of animals through breeding or genetic manipulation but excludes animals euthanized solely for organ or tissue harvest, as well as all procedures less invasive than what corresponds to a needle stick. In Sweden, the definition is wider and dependent on purpose of use rather than level of invasiveness or suffering [1] (Chapter 1: Sections 3 and 4). If the purpose of use is gained knowledge through scientific research, disease diagnosis, drug development, teaching, or other purposes of similar nature, it is considered animal research by Swedish standards, and the animals being used are deemed research animals. There is no requirement of level of invasiveness, and animals for instance observed through behavioral studies, euthanized solely for their organs, or captured and/or killed for species conservation purposes or wildlife studies are included in the definition [1] (Chapter 1: Sections 3 and 4). In the following, we mainly refer to the Directive for general issues, since this is the basis of the Swedish legislation, but also since the Swedish text is not understandable to all readers.

Throughout the EU member states, 9,581,741 animals were used in research during 2017 [2]. In Sweden, the most recent statistics are from 2018 and show that 274,655 animals were used that year according to the European definition and 5,801,463 were used that year according to the broader Swedish definition, including test fishing [3].

Although the definitions may differ in detail, one pillar upon which they rest remains the same: animals have intrinsic value that should be respected and they should be treated as sentient beings [4] (Article 2 p. 21), [5] (Recital 12), [1] (Chapter: 1 Section 1). According to the globally adopted “Five Freedoms”, animals under human care should be spared the negative emotional or physical states of: hunger, malnutrition, and thirst; fear and distress; discomfort; pain, injury, and disease; as well as the inability to express natural behavior [6]. The knowledge that many animal species feel both positive and negative emotions is central to the animal welfare movement and the regulation of animal welfare through laws and guidelines [7]. However, even though the five freedoms and animal welfare legislation aim to prevent animal suffering, animals can be subjected to pain and potential suffering if deemed necessary, for example in animal research. The Directive states (Article 3 p. 1), and has its parallel in Swedish legislation [1] (Section 36 p. 2), that animal research is justified only if ethically approved by a competent authority. However, some animal research, such as observing wild animals from afar, is exempt from this requirement [8] (Chapter 2: Sections 17–25). Pain, suffering, distress, or lasting harm that cannot be considered necessary is not permitted by the Directive (Article 24 p. 2a). Furthermore, the scientific quality in terms of likelihood of success shall be assessed, in order to ensure the relevance, i.e., benefit, of the procedure (Article 38 p. 1a, 2a, 3a). Hence, it is the task of appointed authorities within the member states to determine which research can be condoned and which cannot (Article 36, 38, 40 p. 1a). It is worth mentioning that expressions of values and ethical stances found in the so-called recitals in the Directive are not in themselves legally binding but serve as guidance and ethical background on which the legal requirements of the Directive are based. In the following, we refer to both recitals and articles for the sake of clarity.

### 1.1. Ethical Review of Animal Research

Animal research encompasses many societal and ethical issues and ideals, and there is a global widespread consensus that harm done to animals in the name of science should be regulated and undergo ethical approval before being performed [9]. Within the EU, the Directive sets the ultimate goal of full replacement of procedures on live animals (Recital 10). However, until this goal becomes reality, an impartial ethical project evaluation has to


-Include a full harm–benefit analysis (HBA) (Recital 39, Article 38 p. 2d);-Address the 3Rs (the principles of Replace, Reduce and Refine) (Recital 38, Article 38 p. 2b);-Determine the degree of severity (Recital 22, Article 15 p. 1); and-Ensure that no projects are approved beyond an upper threshold of suffering (Recital 23, Article 15 p. 2).


The Directive further emphasizes consideration of ethical concerns of the general public (Recital 12) and transparency within the review process (Article 38 p. 4). In Sweden, the contents of the directive have been implemented into national animal welfare legislation (the Animal Welfare Act and the Animal Welfare Ordinance) and a detailed guidance document (SJVFS 2019:9/SJVFS 2012:26), known as the L150, from the Swedish Board of Agriculture (SBA) concerning the use of animals for research purposes. An investigation by the Swedish government was in 2011 established to propose a strategy for national consideration of the 3Rs within the sector of alternative research methods, alongside the implementation of the then newly drafted Directive [10]. Furthermore, in 2017, the Swedish 3Rs Center was founded to function as a resource for knowledge and progress concerning the 3Rs and with the goal of minimizing animal use and suffering in research in Sweden.

### 1.2. Past Critique of the Ethical Review

However, fulfilling the aims of the ethical review has proved challenging, and inadequate ethical reviews are a global concern [11,12,13]. Problems brought to light by these international studies can be categorized into two main issues. The first concerns the content of the ethical review process and the way in which it is conducted, including the inherent difficulty in weighing harms and benefits [14,15,16,17,18] and struggles ensuring that the 3Rs are properly taken into account [19,20,21]. The second issue concerns the transparency and legality of how the process is documented and presented, and the unavoidable subjective elements of the weighing process, which have been discussed as a source of uncertainty [16,22,23].

### 1.3. The Ethical Review System in Sweden

In Sweden, as in many other countries, ethical project evaluations are conducted by animal ethics committees (AECs). Sweden’s first AECs were established by researchers in 1979, and their decisions have been legally binding since 1989. There are six regional committees in Sweden today, each comprised of 14 members, whereof six are researchers or research technicians and six are lay persons to enable public insight. At least one lay person, and maximum two, should represent an animal welfare organization. Additionally, to ensure legal certainty, the chairperson and vice chairperson of each committee are former or practicing lawyers.

For this study, we have chosen to analyze the Swedish ethical review process for several reasons. First, multiple previous studies have shown that Swedish AECs may not be as efficient and reliable as intended and that the Swedish method for ethical review may in itself be flawed and difficult to apply [17,24,25,26,27,28]. Second, with a long tradition of including lay persons in their ethical review committees, Sweden makes for an interesting case of the connection between scientific research and public trust. Furthermore, the Directive has been in place for seven years, and the member states are expected to have adjusted well to its framework and requirements. Finally, neither the structure nor the task of the Swedish committees has been revised for many years, and there is a need for insight into how the Directive has been implemented into the work of the Swedish AECs.

Based on the requirements set in the Directive, the above-mentioned challenges of conducting an ethical review, and the current international discourse on what a proper HBA entails and the core role of the 3Rs, we have chosen to put special emphasis on the following dimensions:


-The amount and quality of information being provided by applicants-HBA performance-The 3Rs-The cluster of humane end-points, severity assessment, and the “upper limit” of suffering-The connection between ethical review and public trust


Therefore, these dimensions will be introduced in said order in the sections to follow.

### 1.4. The Importance of Information

For the appointed authorities to fulfill their task of ethical review, the information provided by applicants needs to be complete, correct, current, and relevant [29]. Without the necessary information on which to base their discussions and decisions, the committees simply cannot do what they have been assigned. A report from the Swedish Board of Agriculture [10] addresses the concern (amongst others) of insufficient information in applications resulting in difficulties ensuring the requirements of the 3Rs are met. Already before the implementation of the Directive, poor application content was argued to cause the Swedish AECs to base decisions on insufficient grounds [30]. A study of how the AECs reviewed proposals concerning the use or creation of genetically modified animals further supported this statement. The applications often lacked information of the animals’ situation and yet they were reviewed, and often approved, by the AECs [31]. Correspondingly, problems with project applications observed in European member states include failing to adequately explain benefits, failing to sufficiently address likelihood of success, failing to sufficiently address the application of the 3Rs, and failing to adequately estimate harms [29] (p. 7–8).

### 1.5. Harm–Benefit Analysis

All project evaluations within the EU must contain a harm–benefit analysis aiming at approving only projects where the expected benefit, for humans, animals, or the environment, is in balance with the estimated harm to the research animals [5] (Recital 39, Article 38 p. 2d), [32]. As defined by the Expert Working Group for Project Evaluation and Retrospective Assessment established by the Commission in 2013, a proper HBA requires “a good understanding of the nature and impact of the potential benefits, of all of the expected harms to the animals, taking into account all Refinement measures, and the likelihood of achieving the projected benefits.” [29] (p. 4). The same document further specifies the following guidelines for those performing the review:


-“Should not automatically assume that claims of potential scientific benefit are always correct;-Should understand all the potential harms to the animals;-Should be prepared to challenge the status quo and to reject poorly designed and ill thought through projects and-Be prepared to challenge cultural/social/political issues e.g., outdated methodologies or views that animals do not need pain relief.”


The problems associated with conducting HBAs have been discussed for some time, and ethical advice for addressing its complexity is surprisingly hard to come by [14]. The FELASA Working Group on Ethical Evaluation of Animal Experiments, wrote in 2005 that, “[T]here can be no straightforward ‘algorithm’ for ethical weighing, nor any other quantitative approach that can remove the need for sensitive ethical judgment” [33]. The vagueness of existing guidelines has been welcomed by some authors due to its allowance for interpretation, whilst others have resolved to provide more explicit guidance [14]. In addition to a “heat map model” suggested by the FELASA Working Group on harm–benefit analysis [32], a number of different models for HBA have been proposed over the years in attempts to provide framework or guidance. Among the more renowned worth mentioning are the algorithm models suggested by Mellor and Reid in 1994 [34] (revised in 2004 [35]), Stafleu et al. in 1999 [36], and graphic models such as the “Bateson square” and “Bateson cube” [37,38]. A less well known, albeit more recent, model is Ringblom et al.’s weighting of ethical costs [16].

Several years before the implementation of the Directive, the Swedish AECs were considered to be approving applications too light-heartedly and that ethical debate was not given enough space within the ethical review process [30]. A framework for ethical discussion was proposed as well as suggestions of changes to be made to the Swedish Animal Welfare Act and Animal Welfare Ordinance. In 2004, 40 participants from nine countries at a Nordic European Workshop on Ethical Evaluation of Animal Experiments concluded that the weighing that needs to be done is in itself problematic as it requires balancing very different entities against one another [39], which is a conclusion several studies have since agreed upon [14,22,40,41,42]. Another problem discussed was the fact that the two entities do not share a common time interval. The harms may be immediate whereas benefits, if at all achieved, may arise at different times in the future [39].

Ethical considerations by Swedish AECs have been deemed too shallow [25], and it has been proposed that committees, in Sweden as well as other countries, may regard information concerning potential benefits as more important than information about the harm and suffering of the animals [23,27]. According to Ideland [26], Swedish AEC members had a tendency to focus on discussing experimental methodology rather than the weighing of harms and benefits during committee meetings. There was also disagreement amongst the interviewees on how to define the ethical questions discussed and who was to gain from the ethical evaluation: the animals, the patients, or science. According to recent studies, Swedish AECs still do not fulfill the inherent criteria of HBA and hence, it is argued, ethical justification of animal experiments is not reached through the current review system [17,43].

### 1.6. Replace, Reduce, Refine

The principles of Replace, Reduce, and Refine were coined by researchers Russell and Burch in 1959 [44] and have since been widely accepted and integrated into the planning, execution, and evaluation of animal research globally. According to the Directive, all EU member states are to promote alternative methods (Article 47 p. 1) and ensure prioritization of the 3Rs (Recital 10, Article 4) by ensuring that sufficient information be provided by researchers upon application (Article 37 p. 1) and that the AECs, or their equivalents, include in their project evaluation an assessment of how Replacement, Reduction, and Refinement requirements have been met (Article 38 p. 2b). An HBA requires the consideration of proposed Refinement measures [29]. A mandatory way of achieving Refinement is through the use of humane end-points whereby euthanasia or amelioration of suffering is carried out as early as possible during a study [5] (Recital 14). Information about the humane end-points chosen is to be specified by the researcher upon application [5] (Article 37 p. 1c).

Critique aimed at the implementation of the 3Rs is nothing new. In 1998, a Swedish government-sanctioned report on the use of animals in research concluded that the AECs lacked the competence and resources to determine themselves if alternative methods for proposed research projects were available or not [45]. This left the AECs reliant on information provided by the applicants, which, according to the report, meant that the project evaluation was as such based on insufficient grounds. The report concluded that this dependability would passivate the AECs, jeopardizing the goal of reducing the use of animals in research. Four years later, critique remained and the AECs were criticized of approving projects without having enough information to back their decisions, and the suggestion was made that the AECs should be able to set higher demands on the information submitted by applicants [30].

The lack of sufficient understanding and implementation of the 3Rs amongst AECs has also been identified globally. Curzer et al. [14] argue that the biggest problem of the 3Rs is that they fail to explain *why* researchers should try to minimize harm to animals and that even though there have been discussion of the application of the 3Rs, very little has been published about their key concepts and principles. Graham [46] found in a survey of U.S. Institutional Animal Care and Use Committees (IACUCs) that the majority of participants believed the search for alternatives is most important for research causing more than slight pain to the animals. However, some researchers thought alternatives were not sought for or simply did not exist. Schuppli and Fraser [19] concluded from a survey of members of Canadian ACEs that the 3Rs were rarely mentioned, although some aspects of the concept were applied. Some of the factors hindering the implementation of the 3Rs were incomplete understanding of the 3Rs, a belief that the researchers themselves apply the 3Rs satisfyingly, and lack of consensus on key issues, for example the nature and moral significance of animal pain and suffering. Houde et al. [21] found in interviews that Canadian AEC members mention the 3Rs but that there was still a lack of an understanding of the principles, and they pointed out a need for emphasis on the 3Rs in the AEC members’ instructional training. In a study analyzing membership of IACUCs at U.S. research institutions, Hansen et al. [47] suggested the inclusion of a larger number of scientists who do not conduct animal research, animal welfare experts, and non-affiliated laypeople in the ethical committees to increase attention to and consideration of the 3Rs.

### 1.7. Humane End-Points, Severity Assessment, and the “Upper Limit” of Suffering

After reviewing the proposed project, including applied Refinement measures such as humane end-points, and the described resulting suffering or harm for each group of animals, it is the AEC’s task to classify any approved project’s level of severity as non-recovery, mild, moderate, or severe [5] (Recital 22, Article 15 p. 1), [1] (Chapter 7: Section 10). Guidelines for the severity assessment are found in Annex VIII of the Directive.

Furthermore, it is stated that any proposed procedures involving “severe pain, suffering or distress that is likely to be long-lasting and cannot be ameliorated” are not to be granted approval unless a member state specifically applies for and is granted exemption from this “upper limit” [5] (Recital 23, Article 15 p. 2, Article 55 p. 3), [48] (Section 41c), [49] (Chapter 7: Section 9). This limit could be seen as a special form of Refinement but without allowing for compromise in the pursuit of research goals. Instead, this restriction is absolute [50]. Furthermore, the Directive states that any violation of the ban where experiments have turned out to cause more suffering than legally allowed are to be reported retroactively by the researchers [51] (Annex II p. 7), [8] (Chapter 13: Sections 1 and 3). Up until the writing of this paper, no exemptions from the “upper limit” have been requested by any member state, and as such, no relaxation of the prohibition has been made [52].

### 1.8. Public Interest and Transparency

Public interest in animal research is generally low [53]. Nevertheless, the public’s views on animal welfare and using animals for human benefit are important influences on how animal research is conducted and what projects are granted approval [54]. In a recent survey of attitudes amongst Europeans toward animal welfare, nearly half of those questioned considered animal welfare to be “the duty to respect all animals”, and 94% of the participants believed it important to protect the welfare of farmed animals [55]. In Sweden, 99% found it important. Despite research animals not being specifically mentioned in the survey, its results indicate a general high regard across all member states for the wellbeing of animals. However, opinions on animal research are often far from unanimous. In a recent investigation of the public’s attitudes toward animal research in Sweden, 55% of participants found animal use for medical research acceptable. If ensured the animals were well cared for and not subjected to unnecessary suffering, acceptance reached 82% [56]. This effect of public opinion varying depending on the circumstances of the research, together with the extensive general interest in animal welfare presented above, indicate the need for deeper and more nuanced ethical reasoning regarding animal research.

The Directive promotes public confidence related to the use of animals for research purposes and calls for transparency within the ethical review process (Recital 22, 41, Article 38 p. 4) and the animal research field as a whole (Recital 4, 36). In Sweden, two-thirds (66%) of people questioned trust that animal research scientists correctly abide by the laws and regulations governing animal research, while one-tenth (11%) claim they have little or very little faith that this is the case [56].

In order to achieve transparency, it is crucial that the basis of the approval or disapproval is clearly motivated, and hence the authority responsible for the ethical review must describe in detail the facts and considerations that led up to the decision made [57]. The current Swedish Administrative Procedure Act [58] states that “[a] decision which can be expected to affect a person’s situation in a way that is not insignificant shall contain a clarifying motivation, unless this is without a doubt unnecessary” (our translation). It further concludes that the motivation must include applied regulations and decisive circumstances [58] (Section 32). At the time during which the documents reviewed in this study were composed, the law had a slightly different formulation albeit similar purpose. According to the Swedish administrative law in use in 2017, if a governmental decision is made that impacts an individual (such as a researcher applying for ethical approval of a planned research project), the reasoning behind said decision should be made available for scrutiny. However, only decisions that go against the interests of a party need motivating without exceptions [59] (Section 20). Importantly though, this does not mean that favorable decisions never require motivating. Contrarily, decisions that can be viewed as presidential or that may benefit one party at the cost of another may also require motivating [60] (p. 244). It is evident that when, how, and why motivations are required by authorities is not always easily decipherable.

## 2. Materials and Methods

The documents to be analyzed were obtained through the district courts of the cities where the six Swedish AECs are located: Stockholm, Uppsala, Umeå, Linköping, Göteborg, and Malmö-Lund. The district courts were contacted by email, and all documents were received in digital form. The documents requested consisted of applications that had been sent in to the AECs during the months of March and October 2017 (with one exception, see the first bullet point below) together with their corresponding decisions. These particular dates were chosen for the analysis for the following reasons:


One member of the research group was appointed committee member of Göteborg AEC in November 2017. Therefore, to rule out bias, no documents analyzed were to have been received by the AECs later than October 2017. Before that date, the member in question had no involvement in the committee’s work. For this reason, documents were requested from Göteborg AEC from March and September, so as to ensure that these had finished being processed when said researcher began their assignment. As the AECs meet once a month, this was deemed an appropriate time margin.Applications and decisions dated after the implementation of Directive 2010/63/EU had to be chosen, and preferably from as recently as possible so as to allow for any initial “breaking-in period” regarding implementation of the Directive to have passed and the AECs to be as comfortable and settled in their assignment as possible.March and October have been shown to be two months of the year during which a large number of applications usually reach the AECs (Karin Gabrielson Morton, Expert and Senior policy advisor at the Swedish Fund for Research without Animal Experiments, personal communication, 12 January 2020), thereby allowing for a greater starting population to select from.


We analyzed three documents per AEC. These were selected through a randomized and blinded process for each AEC, i.e., applications were not grouped in relation to content or any other category. First, the total number of documents deemed appropriate for review from each AEC was counted. Then, each document was assigned a number within that range, and three numbers were randomly selected by a third party for analysis.

Then, the 18 documents selected were analyzed with regard to how well their content responded to legal requirements of the L150 (SJVFS 2012:26). An Excel sheet was used with the legal requirements listed whereby each document was revised to determine if it could be ascertained from the contents of the application or decision that the requirements had been met (Appendix A). Depending on how well the contents fulfilled the requirements, this was categorized as Y (Yes), I (Incomplete/Insufficient/Indeterminable), or N (No). To ensure continuity and transparency of the analysis, a guide of how the requirements were to be judged and categorized was conducted before the review started (Appendix A). For cases where it was unclear how certain criteria put forth by the Directive should be interpreted, the research group detailed their chosen interpretation in the aforementioned guide.

The performance of an HBA by the applicant is encouraged by the PREPARE guidelines but not specified by the Directive or the L150 document [61]. However, it is asked for in the Swedish application form. Hence, this has been included for analysis by the present research group, as it was considered necessary for the interpretation of the rest of the results and conclusions of this study to know the full extent of the information provided for the AECs by the applicants.

The extent of argumentation (or justification) of AEC decisions is, other than the requirement for transparency (Article 38 p. 4), not detailed in the Directive but in administrative legislation. Therefore, we have analyzed the decisions according to the Administrative Procedure Act of 2017 [59] (Section 20) with the guidance of Hellners’s and Malmqvist’s book on the subject [60]. There is a clearly labeled section within the decision template where the AEC’s justification and decision are to be specified, and so, this part of the decision has been analyzed for this particular requirement.

To increase scientific validity, the selected documents were analyzed a second time regarding a selection of legal requirements deemed especially important for the study (concerning HBA, the 3Rs, and scientific and humane end-points) by another scientist. So as to avoid influencing the outcome, this person was not privy to the other’s observations, but they were familiar with animal-based research. Then, the two sets of results were compared, and any dissimilarities were discussed and reviewed again using the aforementioned guide (Appendix A) until consensus was reached. The final results can be seen in full in (Appendix A).

Seeing as this study was based on documents from 2017, their contents have been compared to the legal requirements of legislation and guidelines present and in use at the time of application. Thus, unless stated otherwise, the legal documents and guidelines referenced throughout this paper are the versions valid between September and November 2017. Any subsequent changes of importance for the discussion or conclusions of this paper will be clearly stated and referenced.

## 3. Results

A total of 131 documents were obtained: 13 from Göteborg, 16 from Linköping, 32 from Malmö-Lund, 26 from Stockholm, eight from Umeå, and 36 from Uppsala. The annual registers for 2017 from each AEC were also received from all district courts except from Linköping, who gave no reason despite numerous requests.

All documents were revised upon procurement to guarantee that only those including both applications and decisions were ultimately selected. A total of 13 documents missing pages or judged otherwise incomplete were discarded, in spite of several requests to obtain complete review documentation. Decisions both with and without the “The AECs Decision—Non-Technical Project Summary” form supplied by the SBA were included for analysis. To ensure the applications and decisions would be as similar in structure to each other as possible and thereby comparable despite being randomly selected, only initial applications (in Swedish: *grundansökningar*) were included in the draw. For clarity and scientific validity, an initial application was deemed as an application which the AEC upon review had categorized as belonging to category three or four as defined by SJVFS 2008:19 Section 2a, namely that they were self-standing new applications and not changes to pre-existing approvals. A total of 39 documents were excluded for not fulfilling this criterion. Thus, a total of 78 documents were included in the draw. Out of these, all but one research proposal had been approved by the AECs.

### 3.1. Analysis of the Ethical Review

For the sake of clarity, the results from the 18 closely analyzed documents are presented separately for the analyses of applications and decisions. The results for the applications are divided into those obtained from the main “technical body” of the application followed by those from the non-technical project summary (NTPS). For the sake of clarity, the order in which the legal requirements observed are presented here differs from the order in which they appear in the L150 (SJVFS 2012:26) and Excel sheet. Instead, they have been regrouped to facilitate for the reader.

### 3.2. Analysis of Applications

#### 3.2.1. The Main “Technical” Body: Content Provided by the Applicants

A description and motivation of scientific end-points were satisfactorily provided (Y) in all but one application, which was instead judged as I (inadequate) for this criteria.

Description and motivation of humane end-points were included satisfactorily (Y) in 12 out of 18 applications, judged insufficient (I) in five, and completely lacking (N) in one. The demand for clear criteria of when said humane end-points are to be considered reached was fulfilled (Y) in nine out of 18 applications, whilst criteria were deemed indeterminable or insufficient (I) in seven and not provided at all (N) in two.

How the animals’ pain, discomfort, or other suffering is to be observed and determined was only sufficiently described and motivated (Y) in three out of 18 applications. In six out of 18, it was judged insufficient (I) and it was not mentioned at all (N) in half of the applications (nine out of 18).

Five out of 18 applications clearly described the need for monitoring of the animals (Y), while seven out of 18 were insufficient in their descriptions and/or motivations (I), and six applications failed to mention it at all (N).

The application form further requests that the applicant provides an account of how they have reflected when deciding that the benefit of their proposed study surmounts the harm inflicted on the animals: in essence, their HBA. For practical reasons, we have divided this one point in the application form into two, judging the mention and deliberation of benefit and harm separately. In seven out of 18 applications information on harm was judged satisfactory (Y), whilst it was unsatisfactory (I) in five and not mentioned at all (N) in six. The reasoning around benefit was deemed satisfactory (Y) in nine applications, unsatisfactory (I) in six, and not included (N) in the remaining three.

#### 3.2.2. The Non-Technical Project Summary: Content Provided by the Applicants

Certain legal requirements specifically regard the NTPS section of the application. For example, it is within the frames of the NTPS that the researcher is requested to describe their application of the 3Rs. For two of the 18 applications, a non-technical project summary was not included at all. Hence, it should be kept in mind that these will represent two of the N for each of the requirements below.

The non-technical project summary (NTPS) should state the number and type of animals requested for use. In 16 out of 18 applications, this was done as required (Y).

Within the realms of the NTPS, the applicant is further required to inform about the aim and benefit of the project as well as the suffering of the animals. In the vast majority of all applications (16 out of 18), the aim of the proposed project was well defined (Y). Half of all analyzed applications satisfactorily informed about the benefit of the project (Y), seven informed but unsatisfactorily (I), and two lacking an NTPS did not share any information of this at all (N). Regarding the harm done to the animals, seven out of 18 applications gave sufficient information (Y), another seven gave insufficient information (I), and the remaining four did not inform about the harm at all (N).

Regarding the 3Rs, half (nine out of 18) of the applications contained satisfactory accounts (Y) of how Replace had been achieved, whilst seven only gave unclear or insufficient accounts (I), and two failed to mention this at all (N). For Reduce, seven applications contained satisfactory accounts (Y), eight contained unsatisfactory accounts (I), and three had no information regarding this at all (N). Similarly, seven applications gave satisfactory information (Y) about Refine, six gave insufficient information (I), and five gave no information at all. There were no patterns indicating that the satisfactory inclusion of one R in an application was consistent with the other Rs having been deemed satisfactory for said application as well, nor that applications failing to account for one R would do so to a greater extent for the other Rs, too.

Furthermore, Reduce and Refine were occasionally confused with each other by the applicants. In three out of the 15 applications describing Reduce (graded as Y or I), Reduce was described under the Refine section, and in two out of 13 applications (likewise graded Y or I), Refine was described under the Reduce section. Replace was, if graded as Y or I, described under the correct section throughout.

#### 3.2.3. The Non-Technical Project Summary: Content Provided by the AECs

The AEC is required to amend the NTPS submitted by the applicant with a definite degree of severity, any changes or additions needed for project approval, and a detailed account of whether the proposed project is to be subject to retrospective assessment or not. This can, but must not, be performed in a template designed for this purpose. The results were the same for all of these three requirements respectively. Four out of 18 applications were categorized as Y, none were categorized as I, and the remaining 14 were categorized as N. All decisions marked N were so labeled as they did not contain the form “The AECs Decision—Non-Technical Project Summary” nor any other document indicating that the criteria listed above had been met.

### 3.3. Analysis of Decisions

In a vast majority of the decisions (16 out of 18), the principal investigator and director were specified (Y). The remaining two, both from the same AEC, failed to disclose this information altogether (N). In addition to this, the same decisions by the same AEC also failed to disclose the location of the proposed study (N), as did the third decision by said AEC. All remaining 15 decisions from other AECs contained the location information (Y).

All 18 decisions contained a specific section labeled “Terms of decision” and were thereby, regardless of what terms were listed therein, considered to live up to the requirement to disclose the terms of approval specified by the AECs (Y). Similarly, all 18 decisions contained a final assessment of the level of severity (Y).

To analyze the presence, or lack thereof, of differences of opinion (meaning that one or more of the committee members disagree with the committee decision), the categorization of Y, I, and N had to be carefully defined and differed somewhat from the demands set for other requirements (see Appendix A). Nonetheless, one decision contained a difference of opinion clearly stated (Y). Three decisions, all from the same AEC (not the AEC mentioned in the first paragraph of this section), referred to an appendix of the decision protocol for differences of opinion but did not include this in the documents initially requested by the research group, whereby these were marked as I. The district court did not supply the protocols at a later point of the study despite being asked specifically and repeatedly to do so. The remaining decisions did not include or indicate the presence of any differences of opinion (N).

Replace, Reduce, or Refine were not included in any of the 18 decisions analyzed. Consequently, all 18 decisions failed to fulfill this criteria (N).

As for the applications, harm and benefit in the decisions were judged separately. In four out of 18 decisions, harm was satisfactorily mentioned and described (Y), whilst in another four decisions, harm was not mentioned at all (N). The remaining 10 decisions simply included a boilerplate stating that, “The committee considers the importance of the project to outweigh the suffering of the animals” (our translation). This was marked as insufficient (I), as it was considered a statement of the outcome of the decision rather than the reasoning behind the weighing of harms and benefits. Likewise, four out of 18 decisions satisfactorily mentioned and described the benefit (Y), four did not mention it at all (N), and the remaining 10 included only the aforementioned boilerplate (I). The decisions that fulfilled the criteria (Y) for the description of harm were also the ones that fulfilled the criteria (Y) for benefit. The same correlation was true for the decisions that failed to describe harm or benefit.

Out of the 18 decisions analyzed, four satisfactorily motivated the decisions in accordance with administrative law (Y), whilst three included no motivation at all (N). Three out of the four satisfactory motivations were provided by the same AEC (meaning that all reviewed decisions by said AEC fulfilled this criteria), and the fourth decision provided by another AEC was judged satisfactory due to the inclusion of terms in the approval and as such there being more information given than the standard boilerplate. Out of the remaining 11 (I), eight only included the aforementioned standard boilerplate, whilst three did not contain a motivation per se but a reference to the review panel documents. The review panels are to submit a written and motivated proposition, or if more appropriate, a written decision basis for the AEC to base their final decision on. In 16 out of 18 cases, a written document was produced and submitted (Y), and in two, there was no record of such (N). Half of the 16 written documents (eight out of 16) included a satisfactory motivation (Y), one included a brief standard motivation deemed unsatisfactory/incomplete (I), and seven contained no motivation at all (N).

### 3.4. Additional Results

Based on the material obtained for the study, we gained results outside the pre-established foci, whereof themes relevant for a more comprehensive understanding of the results presented above are presented in the following.

#### 3.4.1. Discrepancy between Law and Application Form

When creating the Excel sheet used for the analysis, it became clear that there is a discrepancy between the demands set by legislation and the questions asked in the application form. For example, that the applicant accounts for *how* the animals’ pain, discomfort, or other suffering is to be observed and determined is a specific requirement detailed in the L150 [62] to be considered, described, and motivated. However, even if a description of the potential impact on animal welfare is requested by the form, there is no question or section in the application form regarding method for welfare assessment or scoring of potential impairments (aka the *how* in question). Similarly, the form does not include a request for information about the specific judgment criteria for humane end-points, nor the consideration, description, and motivation of the need for monitoring—although this is requested by the L150 [62]. In addition, as mentioned above, how the applicant has taken the 3Rs into consideration is only given specific space to be described within the NTPS, and not asked for elsewhere by the form, i.e., risking omitting details relevant to the AEC. In neither the Directive nor the L150 [62] is this placement referenced. Thus, there are gaps between the requirements of the legislation and the information requested in the application form in several points related to core values of the ethical evaluation and decision-making process.

#### 3.4.2. Inconsistencies in Documentation and Archiving of Documents

As stated in Article 45 of the Directive, “all relevant documentation” concerning the ethical review should be kept for at least three years from the expiry date of the project authorization. All AECs seemingly fulfill this time frame criteria at a first glance, as the research group was in the spring of 2020 able to request and receive documents from 2017. However, exactly what documents had been saved and were made available for our study differed between committees. For example, in only two out of the six committees was the form “The AECs Decision—Non-Technical Project Summary” included amongst the saved documents. This form is specifically tailored by the SBA to enable the committee to add to the NTPS a degree of severity, requested changes (if any), and the possible need for retrospective assessment. Since the form was not included and as such seemingly not used by the other committees, they were judged as failing to live up to these demands because it was not ascertainable whether or not they had revised the NTPS. As no other documentation was provided to show that revision had taken place, the legal requirements were not met. Furthermore, the filing order of the different sections of the documents was inconsistent both between AECs and between handled cases within one and the same AEC. Early and final versions of applications, additions, or alterations made per email, decisions to postpone or to approve, review panel proposals, and the few “AECs Decision—Non-technical project summaries” received by the research group were inconsistently arranged by the different district courts. Due to this, a thorough review was needed in order to know for sure whether the documents received were in fact complete. As demonstrated by the need for an initial review of the documents described under “Methods” above, individual or multiple pages had on occasion been left out, rendering these documents useless for analysis. One district court was asked by the research group to provide the protocols from two AEC meetings as references had been made in decisions that differences of opinion could be found there. However, the district court did not grant this request and gave no reason. Therefore, it is unclear if said notes were saved or not, assuming that the AECs had kept minutes as stated and legally required [62] (Chapter 7: Sections 12 and 13).

#### 3.4.3. Confusion Around the “Upper Limit” of Suffering

During the study, we came across the discussion of how the “upper limit” of suffering [5] (Article 15 p. 2), [48] (Section 41c), [49] (Chapter 7: Section 9) should be interpreted and applied. Statistical documents as well as correspondence with the National Animal Ethics Committee (NAEC) revealed that Sweden up until the writing of this article had not once handled a case where a project had been considered to exceed the upper limit [3].

The Working document on a severity assessment framework supplied by the Commission [63] includes only one brief mention of the upper limit. The document provides an example of a study of severe long-lasting suffering caused by arthritis in mice, which could be “authorised subject to an otherwise positive project evaluation including review of the harms and benefits” [63]. Apart from this example whereby “a number of weeks” is evidently considered prolonged (for this particular scenario), no definition or precedent of what constitutes “long-lasting” or “severe” with regard to Article 15 p. 2 exist as far as the research group have been able to find out. Nor is “cannot be ameliorated” defined and so to what degree the suffering has to be lessened is unclear.

#### 3.4.4. Unclear Rules of Motivation of Authoritarian Decisions and Transparency Requirements

As mentioned in the introduction, the reasoning behind decisions made by authorities should be made clear, at least when the authority has ruled against the interests of a party, the outcome benefits one party at the cost of another, or when the decision could be considered especially important for the ruling of future cases of similar nature [59] (Section 20), [60] (p. 244). However, exactly how this should be applied to the decisions made by AECs seems unclear, since there was no consistency amongst the documented decisions analyzed in our study as to what extent a motivation was included or what it should contain. In addition, the SBA has presented no guidelines regarding this point. Correspondingly, there exists no clear definition or guidelines determining how the transparency required by the Directive should be achieved.

## 4. Discussion

### 4.1. Harm–Benefit Analysis

In order for an HBA to be performed correctly and live up to its purpose, all harm needs to be known by its performers [23], [29] (p.4). Correspondingly, so do the benefits of the proposed research and the probability of their occurrence [23,64,65,66]. To aid in assessing the level of harm of a project, the reviewing authorities may look to the classification scheme in Annex VIII of the Directive. However, the Directive does not provide guidance for assessing proposed benefits [42]. As it was already in 2004 found that costs (today “harms”) were in general easier to identify and define than benefits [39], this is surprising. However, contrarily, in human clinical studies, harm has instead been regarded as the more difficult of the two to identify [23]. To our knowledge, the reason for this difference is unknown. One way to facilitate the weighing of harms and benefits could be to develop and ensure the use of proper professional terminology for benefits similar to that which already exists for harm, as suggested by Brønstad et al. [23]. This would avoid discussions of benefits from becoming too general and allow for a more nuanced and exhaustive ethical debate. The risk of the HBA being constrained by “situated ethics”, whereby ethics is viewed as something that is contextually and situationally negotiable [26], would also decrease if the committee members use the same language, so to speak.

In addition, a further point of discussion lies inherent in the requirement to perform an HBA, namely how to handle the general claim of performing an ethical evaluation, which can be regarded to either cover all but only measurable points usually considered in an HBA, or, on the other hand, understood as an expectation to build the ethical analysis on a wider understanding of harms and benefits. This can at times be expressed as a need to go beyond the HBA to ensure that a proper ethical evaluation is performed. As put forth by Alzmann [22], a weighing of competing interests can only be carried out if all aspects relevant for the case are considered, and so, a comprehensive analysis of considerations should always precede the weighing. The parties responsible for assessing the ethics of a proposed research project should according to Alzmann “employ a comprehensive and thorough approach that includes but is broader than harm–benefit analysis” [22]. Correspondingly, Röcklinsberg et al. [27] concluded that the authorities responsible for conducting the ethical review must widen the aspects included in the decision-making process to be able to live up to the intent of the legislation. Alzmann [22] further argues that there are ethically relevant aspects beyond the procedural suffering, pain, distress, and harm of the animals and the benefits for humans, animals, or the environment as formulated by the Directive, and that there is need for the development of a catalogue also encompassing such things as husbandry conditions and non-physical suffering. Since the HBA should take all harms into account, it is reasonable that the possible pain, stress, or suffering caused not only by the procedures themselves but also by handling of the animals, cage conditions, social isolation, gene modification, and other parts of husbandry be considered. A number of catalogues have been suggested, and all include the same and/or similar criteria to be evaluated, but a catalogue that is both practical and exhaustive is yet to be developed [22].

Furthermore, it is important to note that harm or suffering does not have to be physical [14]. On the contrary, the Directive repeatedly emphasizes “distress” in animals and the definition of suffering provided by the L150 (Chapter: 1 Section 8) clearly states that psychological harm or suffering is included in the concept [8,62]. Despite this, we found it rarely mentioned by the applicants and seemingly often overlooked by the AECs.

According to the PREPARE (Planning Research and Experimental Procedures on Animals: Recommendations for Excellence) guidelines [61], the researcher should perform a harm–benefit assessment and justify any likely animal harm. However, since our study shows that the harms were not always properly accounted for by the applicants and applications have been approved nonetheless, this indicates that the AECs might not have the knowledge or guidance as to what constitutes the concept of harm and what information they should be receiving from the applicants. Additionally and likewise alarmingly, seeing as the manner in which the ethical weighing is performed is of great importance [22], studies have shown that how harms versus benefits are balanced is inconsistent and that there are discrepancies amongst the performers as to which benefits may justify which harms [26,67]. As a consequence of the lack of visible HBAs in the decisions we have analyzed, our study can neither confirm nor deny that this might be occurring amongst Swedish AECs. Seeing as the present review system in Sweden can be argued as failing to achieve ethical justification for animal research [17,43] and applications are continuously approved on insufficient grounds, as revealed by the results of our study, the need for further review of the process and suggestions for how to improve and facilitate the HBA process is pressing.

One may debate if the inclusion of ethical reasoning by the applicant is solely positive or if it may in fact also be counterproductive. If the applying researcher includes an ethical reasoning and it is viewed as comprehensive enough by the AEC, as was the case in the majority of applications in our study both regarding harm and benefit, this on the one hand proves that the applicant is aware of existing ethical dilemmas, is able to reason around them, and reach a conclusion of ethical justification. However, on the other hand, if the AEC is too reliant on the information given and too readily accepts the applicant’s arguments and viewpoints as true, the committee is at risk of passivation of a similar kind to that discussed in 1998 [45]. The Swedish AECs were already almost two decades ago criticized for not placing enough emphasis on the ethical debate [30]. Hence, we argue that there is a risk that they may base their own ethical reasoning and justification off of the ethical standpoint of the applicant. In line with the urging by the Expert Working Group for Project Evaluation and Retrospective Assessment [29] (p. 4) that the AECs “should not automatically assume that claims of potential scientific benefit are always correct”, we argue that they should avoid accepting the ethical reasoning of the applying researchers too light-heartedly. The inclusion of a full HBA by the AECs in the decisions of project evaluation should be self-evident. However, the theory of passivity is contradicted in our study by the fact that the AEC, which had included satisfactory HBAs in all their decisions, had reviewed applications containing generally very satisfactory ethical reasoning by the applicants.

Even if all the needed information for the HBA is supplied, the weighing process in itself is far from straightforward, and the ethics involved in balancing harms versus benefits are not clearly identified [14]. According to Schuppli [67], neither harms nor benefits are really quantifiable nor in commensurable units, which is highly problematic. To balance interests against each other, one has to consider the principle of proportionality whereby the weight or significance of the conflicting values is determined so that one may know how to balance the scales [42]. However, the weighing of harms and benefits in animal research can, as previously mentioned, be likened to the comparing of apples and oranges [14,22,23,39]. Within the ethical review of animal research, non-human interests have to be dealt with in a legal context more or less solely catering to the interests of humans [14,42]. That is, the animals’ situation is never in its raw form truly comparable to the other side, as non-humans do not have the same legal status as humans [40] and, as emphasized by Orlans, almost all the harm is on the animals’ side whilst all the benefit is on the humans’ [68]. As animals are not seen as worthy of equal consideration to humans, the weighing of animal interests against human interests is troublesome [40]. Not only does the weighing thereby metaphorically concern different kinds of fruit, it is arguably inherently biased, as an apple will always weigh less than an orange. Furthermore, HBA means balancing harms that are more or less guaranteed, perhaps not regarding their extent but their occurrence, against benefits that may well be anticipated but cannot truly be defined until after the experiments have been conducted [66]. Similarly, harms will occur at certain calculated moments and last for fixed periods of time, whilst benefits may occur at unknown points in the future or never at all [39]. This imbalance is most relevant when assessing proposals of basic research seeing as possible benefits of the acquired knowledge are unknown until the research has ended [42,66]. Curzer et al. [14] illuminated the difficulties of classifying knowledge as a benefit by raising questions such as “What counts as knowledge?”, “How are different bits of knowledge to be compared?” and “How are the bits of knowledge interconnected?”. They argue that the meaning of knowledge is as vague as that of harm, and that it is inevitable that different appointed authorities will evaluate the terms differently from each other.

Another argument put forth against the usefulness of the HBA is that placing an absolute cap on the amount of suffering allowed contradicts the consequentialist system of harm–benefit analysis as a tool for ethical evaluation [50]. Although, theoretically, the concept of HBA should mean that any amount of suffering could be outweighed by even greater benefits, we do not agree that this is a valid argument for abandoning the use of HBA altogether because: (a) if clearly defined and harmoniously interpreted, the “upper limit” should not pose a threat to the weighing of harms and benefits but rather work as a beneficial complement to improve Refinement; (b) some actions simply cannot be justified no matter the benefits reaped due to the grave moral or ethical consequences (such as public distain). However, the question of whether an “upper limit” can be applied to the amount of harm/suffering allowed matters little if the process for ethical review in itself does not work (as demonstrated above). For example, if the AECs are not provided access to all aspects of harm, they will not be able to determine the degree of severity and will hence for especially invasive procedures not be able to tell if the upper limit should in fact be considered breached and an HBA not be performed at all. Therefore, it is vital that the ethical review process, including methods for HBA and applicability of the upper limit, is thoroughly revised sooner rather than later and preferably on an international level.

It is unknown from our study if the AECs that did not include HBAs failed to do so due to ignorance, the aforementioned possible passivity, lack of knowledge about what was required of them with regard to transparency, or perhaps as a direct result of not having included sufficient ethical discussion of harms and benefits during the reviewing process. To know this, further research is needed, perhaps including in-depth surveys or interviews of the AEC members and/or auscultation of committee meetings.

### 4.2. Replace, Reduce, Refine

#### 4.2.1. Content Provided by the Applicants

Not more than half of the applications included satisfactory descriptions on how the different Rs had been achieved (within the NTPS where this was specifically requested). This result suggests that the applying researchers inadequately considered the 3Rs throughout the planning of their projects and hereby did not fulfill the intentions of the legislation. There may be several explanations for this.

First, the applicants may not have enough knowledge of the 3Rs and how to apply these principles throughout the research process, or perhaps they underestimate the importance of their inclusion. The ultimate goal of the EU Directive is a complete replacement of research animals. In line with this, Curzer et al. [14] and Franco et al. [69] stressed the importance of the hierarchal application of the 3Rs proposed by Russell and Burch [44] where first priority should be to Replace, followed by Reduce, and lastly Refine. Animal researchers have an overall good knowledge and a positive attitude toward the 3Rs [70,71,72,73]. The 3Rs should be considered throughout the entire research process, including application writing and ethical review [74] and according to Curzer et al. [14], the “should” expresses a moral obligation of researchers to minimize harm to animals. In a survey amongst Dutch researchers in 2011, the majority of the 46 respondents answered that the 3Rs play a role in the application process [72]. Approximately half emphasized Replacement, while Refinement and Reduction were considered relevant by nine out of 10. Schuppli and Fraser [19] found that the majority of questioned Canadian researchers considered the 3Rs equally important. More recent studies show that researchers seem to be least in favor of Replacement whilst finding Refinement the most feasible or important to apply [69,72]. However, our results do not seem to mirror this attitude, as half of the applications contained satisfactory accounts of Replacement and approximately one-third described Refinement satisfactorily. However, to be able to say with scientific certainty if such is truly the case, a larger sample population is needed. In addition, the possible impact on the results of the 3Rs being confused with one another by some researchers, as seen in our study as well as others, would need to be investigated [70,73]. Interestingly, in our study, these mix-ups have not been corrected by the AECs in the documentation of their decisions. Schuppli and Fraser [19] suggested that Refinement may be difficult to understand because of its various applications. Proper definitions are crucial for an appropriate review, and these findings highlight the need for increased understanding of the 3Rs among both applicants and AECs. In what way knowledge and attitude among Swedish researchers today may affect the description of the 3Rs in the AEC applications warrants further research.

Second, the applicants may not be aware of how to describe the application of the 3Rs in their research. Guidelines on how to plan research with animal use, such as PREPARE [61], can aid both the application and ethical review process, and thorough planning by researchers can assist AECs in their assessments of research projects. This will contribute to less wasteful use of animals, which is an important part of the implementation of the 3Rs [61,74,75]. However, if the researchers are not adequately guided as to how to provide the needed information to the AECs, or what constitutes said information, for example through application forms not corresponding to legal demands, their task is not an easy one.

Third, the NTPS provides information adapted to the layman, and the applicant may therefore intentionally not include a detailed, technical description of the 3Rs in this section. Hence details are instead often spread out under different headings in the “technical body” of the application form, however without a clear reference to the 3Rs. This leads to a more difficult evaluation by the AEC and challenges the validity of the ethical review as the AEC risks missing important information. As studies have shown, the AECs are often short on administrative staff for the preparation of documents as well as meeting time, and hence inconsistencies as to where in the applications information of certain importance may be found can only be expected to strain their resources further.

#### 4.2.2. Content Provided by the AECs

The AECs are responsible for ensuring that the 3Rs have been duly considered and applied throughout the proposed study by the applying researchers. Varga [41] suggested that the prevention of harm, i.e., Refinement, is the most important duty of the AEC and that animal suffering is a potential measure for outcome assessment of the performance of AECs. However, as previously mentioned, the AECs cannot fulfill their task without access to the information on which they are to base their decisions, and as shown, this information is sometimes lacking or scattered due to the structure of the application form, which increases the difficulty of gaining a good overview of the researcher’s account of harm and the 3Rs. The AECs need to understand the inflicted harm, i.e., having an understanding of the 3Rs, particularly Refinement [29] (p. 4). The unsatisfactory descriptions of harm in the HBAs, together with omission of the 3Rs in the decisions by the Swedish AECs included in our study, give the impression that in general, this may often not be the case. However, in a previous study, Hagelin et al. [76] found that the majority of modifications of applications requested by Swedish AECs were in fact related to Refinement. Furthermore, it is clear that Refinement and Reduction often go hand in hand [77], but in some circumstances, Refinement measures minimizing animal suffering may only be only feasible if a larger number of animals are used [69,78,79,80]. Thus, evaluating the 3Rs is not an easy feat and requires balancing the Rs against each other. Nonetheless, it needs to be done. Not in any of the decisions analyzed in our study have the AECs mentioned Replace, Reduce, or Refine. Consequently, all 18 decisions failed to fulfill these criteria. Both Houde et al. [20] and Schuppli and Fraser [19] noted that AEC members bring up the 3Rs during AEC meetings, and it cannot be ruled out that this is occurring also in the AECs participating in our study. However, it was not documented in the decisions and the lack of headings and space reserved for the 3Rs in the decision template do not facilitate the inclusion.

It may be possible that the AECs, similar to the applying researchers, may lack the proper understanding of how to apply the 3Rs. Alternatively, the AECs may not be clear as to which extent the 3Rs need to be included in the decisions and HBAs. Schuppli and Fraser [19] found that some AEC members trusted that researchers considered and applied Reduction and Replacement sufficiently themselves, which may hinder the consideration of the 3Rs by the AECs. Incomplete understanding of the 3Rs, lack of consensus on key issues such as the nature and moral significance of animal pain and suffering, were also considered to impede the 3Rs’ implementation. Furthermore, studies have shown that AEC members may have doubts about the applicability of Replacement, e.g., when alternatives are sought for full body models [19,81], creating obstacles for its implementation by the AECs [19]. The authors also described five aspects limiting the discussion of Reduction by AECs during the review: emphasis on sample size rather than experimental design; lack of expertise to critically evaluate numbers of animals; confidence that researchers apply Reduction adequately themselves; confidence in the scientific peer review by granting agencies; and the concern of harm over numbers. The lack of inclusion of the 3Rs within the decision is not only an omission of the 3Rs per se, but it may mirror an inadequate consideration of the 3Rs throughout the review process.

Despite our choice to address them separately in this article, there is no clear line separating the 3Rs from the HBA. Rather, they are very much entwined [14]. The 3Rs shall, in addition to improved animal welfare, provide a high scientific quality with benefits for society. Graham and Prescott [79] gave examples from research using animal models of disease on how appropriate model selection, proper study design, questioning the value of translational research, and using less stressed animals provided more reliable results, ultimately contributing to the anticipated benefits for society. As such, approaching Refinement from a broad perspective that considers the animal’s interest as well as promotes the scientific objective both decreases harm and increases benefit [79]. However, this balance is difficult and requires skills in modern research, animal science, animal welfare, and applied ethics, because defining “unnecessary suffering” depends on the interpretation of a series of complex factors [27]. Thus, if the AECs are discussing technical and medical terms rather than animal welfare and ethical issues [26,27,82], this may be hindered. Clearly, similarly to the applying researchers, the AECs need to not only ensure that the 3Rs have been applied but consider them in their harm analysis, as well as in their benefit analysis.

### 4.3. Humane End-Points, Severity Assessment and the “Upper Limit” of Suffering

Directly linked to the level of harm, the application (or lack thereof) of the 3Rs and HBA are the use of humane end-points, severity assessment, and the “upper limit” of suffering.

The severity assessment “shall be based on the most severe effects likely to be experienced by an individual animal after applying all appropriate refinement techniques”, and additional factors such as humane end-points shall also be taken into account [5] (Annex VIII Section II). The Directive proposes the use of “early and humane end-points” (Recital 14, Article 13 p. 3), and an account of planned humane end-points as a Refinement strategy is explicitly requested by the L150 [62] (Chapter 3: Section 4). Despite this, progress in implementation of humane end-points has been slow in several areas of research concerning severe diseases such as sepsis, cancer, and sclerosis [50]. Amongst the reasons given for this reluctance to change is peer pressure to comply with established norms. However, arguments have also been made that earlier and more humane end-points results in more precise scientific results as data is collected before the animals observed deteriorate, become moribund, or die [50,79,83]. In our study, humane end-points were only satisfyingly described and motivated in two-thirds of analyzed applications, and clear criteria for end-points were provided for just three quarters of those. Hence, not all applicants fulfill their task of completing the form as requested, and in turn, the AECs determine levels of severity on the basis of insufficient knowledge about the animals’ situation. Even though the reasons behind the lack of sufficiently described humane end-points are unclear and there may be several, the consequences are obvious and grave. Humane end-points are a mandatory part of Refinement and, as previously mentioned, of great importance both to avoid unnecessary animal suffering and to ensure reliable scientific results. Therefore, their exclusion from the ethical review can be expected to have a negative impact on animal welfare, research validity, and in the long run, public trust in animal research.

When determining the appropriate degree of severity for a proposed project, the Directive states that, “The severity of a procedure shall be determined by the degree of pain, suffering, distress or lasting harm expected to be experienced by an individual animal during the course of the procedure.” (Annex VIII). Thus, being aware of the full extent of the harm inflicted on the animals is pivotal, something we found that the documentation of the AECs decisions could not show was the case. As a result, there is a risk that some degrees of severity determined by the AECs may have proven to have been incorrectly anticipated once the research was conducted. However, to determine if this was in fact the case, one would have to cross-reference AEC decisions with retrospective reviews by the NAEC, which is something we have not been able to do within the confines of this project but would highly recommend be done. However, it is worth noting that only projects of the highest degree of severity are currently mandatorily reviewed by the NAEC and would therefore be the only ones that could be scrutinized in this way. It is our opinion that comparing the decisions with the researchers’ own reporting of “actual level of severity” would not be scientifically just, because there is a risk of bias in the researchers’ reporting. Additionally, as will be further discussed below, we believe the reporting of animal research statistics by researchers is unreliable and in need of revision, specifically concerning severity assessment.

There is an absolute “upper limit” to the amount of pain, suffering, or distress an animal may be subjected to in the name of research [5] (Article 15 p. 2, Article 55 p. 3), [48] (Section 41c). However, what value does a strict “upper limit” hold if those abiding by it do not know how to apply it? Unfortunately, we received no answer from the NAEC or the SBA as to how the AECs are to interpret the “upper limit” in Sweden. However, this is not surprising, as seemingly very little guidance on the subject can be found, and the words “severe”, “long-lasting” and “ameliorated” used by the directive may be interpreted in a number of ways.

Olsson et al. [50] state that “severe” suffering is “more than merely a quantitative increase in negative state” and that it occurs “when negative experiences dominate attention; there is limited capacity for distraction or compensation; normal life cannot be pursued; full recovery cannot occur even if the external situation improves; or (in humans) one’s own life is judged not to be worth living.” They claim that existing methods of assessing animal welfare fail to focus on these qualitative features of severe suffering and that there is need for “insight into how animals are affected by the total load of aversive experiences (including a consideration of additive, multiplicative, and cumulative effects) to which they may or may not habituate.” As such, biomarkers or visible changes such as cortisol levels or bruising are according to Olsson et al. [50] not reliable enough beyond a certain point of suffering. A FELASA Working Group Report by Fentener van Vlissingen et al. [84] similarly conclude that an overall assessment and the duration of a condition needs to be taken into account when determining severity and that severity is comprised of the clinical condition of the animal as one aspect and the procedures it undergoes as another. Suffering involves “complex, subjective experiences” [85]. Thus, we ask the question if commonly used criteria such as weight loss, porphyrin staining, movement and posture deviations, or piloerection could in fact be questioned for use when the proposed projects are expected to cause severe suffering? Furthermore, when severe suffering takes place, pain or stress will in itself interfere with the animal’s ability to store and recall information [50]. Thus, “the ability of the animal to take control and ‘tell’ us anything about its own state becomes limited.” This means that conditioned place preference tests and other behavioral tests of similar nature could be deficient for measuring suffering regarded as severe [50]. Hence, an assessment model taking into consideration the extent to which damage to one functional system, for example caused by great pain, can impact another may be preferable. Violence or other disrupted social behavior, chronic fatigue or sleep loss, reduced appetite, or atrophy of brain regions may all be observable signs that an animal is no longer able to mount an adaptive response or cope with its situation [86]. As shown by Lindl et al. [87], there exists an inverse correlation between the severity of animal research projects and the ultimate applicability and value of their results for humans. Hence, higher levels of severity could arguably, apart from being larger harms themselves, be considered as directly detrimental for the weight of the proposed benefits. Bearing this in mind, we believe it is of outmost importance for a correct ethical evaluation that severity can be reliably assessed at all levels and that the “upper limit” is enforced as intended.

According to Beauchamp and Morton [85], the Directive lacks information about what both intensity and duration of harm constitutes. They argue that these two components require detailed conceptual analysis and that any “upper limit” they are applied to must be both practically measureable and avoidable. When it comes to defining what constitutes “long-lasting”, looking at the term through the lens of human experience and opinion is probably of little help (Paul Flecknell, personal communication, 25 August 2020). Similar to the practice of determining animal sentience based on typically human characteristics or behaviors, there are risks associated with applying an anthropomorphic way of thinking when determining animal suffering and in assuming that what affects humans will have similar impact on other animals [22]. Instead, Flecknell toys with the idea of translating time intervals of pain from man to mouse or vice versa by placing them in relation to the average life span of their species. A similar approach does exist in the Directive where the “lifetime experience” is to be considered when contemplating the reuse of research animals [5] (Preamble 25) but nowhere else. Hypothetically, if a human is in extreme pain for one month of their life, how many days, hours, or minutes would this correspond to in a mouse? The idea of such a method is certainly interesting, albeit difficult if not impossible to apply; however, it does not provide an answer to the question at the base of the issue: How much time or how large a proportion of an individual’s life could be allowed to consist of severe pain or suffering? Especially when said individual is most likely not the one reaping the fruits of its sacrifice. In addition, consideration must be given to the fact that the extent and nature of prior harm may impact the experience of subsequent harm [85]. Flecknell concludes that, based on the Directive and associated documents together with his suggestion of looking at time as a proportion of one’s life span, severe pain or distress that lasts 24–48 h in a mouse should, in his opinion, be seen as long-lasting. However, he does point out that he believes the often automatic assumption that duration increases severity is flawed. Instead, he argues that, “severity is something that requires careful assessment in an individual animal” and that “duration of the adverse effects is something to consider, but not as a reason to automatically increase the severity category.” Finally, he adds that if an animal has little or no sense of “future” but rather “lives in the moment”, then the concept of long-lasting pain would be very different for such an animal, and this should be taken into consideration when estimating duration of suffering.

Although the aforementioned lack of consensus regarding the “upper limit” is hardly surprising, it is nonetheless alarming. Without consensus amongst member states and responsible authorities as well as national AECs (or their equivalents) regarding its interpretation and implementation, the prohibition is all bark and no bite, and its purpose ultimately toothless. A robust and comprehensible prohibition of inflicting suffering above a certain threshold would according to Olsson et al. [50] lead to an increased motivation amongst researchers to find alternative methods. Thus, establishing a firm “upper limit” of suffering and enforcing it would not only spare animals from suffering but encourage application of the 3Rs and, as previously mentioned, possibly result in more reliable scientific results. Limiting the suffering of research animals in this way could also lead to increased public support for animal-based research as severe suffering inflicted on animals, be it for the sake of research or otherwise, is perceived by many as unacceptable [50,88].

As is evident, information about how suffering is to be observed and graded, including clear judgment criteria and/or assessment templates, needs to be provided by the applying researchers. If not, the AECs are not able to correctly ascertain whether or not Refinement strategies have been sufficiently considered nor if there is a danger that the suffering will not be judged correctly, possibly risking exceeding the “upper limit”. Without this information, especially painful or otherwise detrimental projects and procedures risk slipping through the system as “severe” when they should never have been approved at all. Thus, the fact that only 17% of applicants in our study fulfilled the criteria of describing how the animals’ pain, discomfort, or other suffering was to be observed is beneath criticism.

Regrettably, the lack of knowledge as to how and when the “upper limit” should be applied may allow, or already have allowed for, projects to be labeled as “severe” when in fact they may have transgressed the “upper limit” [89]. As a result, the welfare of an unforeseeable number of research animals would be endangered as they risk being subjected to unnecessary and unlawful suffering, directly contradicting the aim of the Directive and the “Five Freedoms”. Furthermore, the AEC would then not only have failed its task as a committee, but such an approval could be seen as in breach with both legislation safeguarding animal welfare as well as the administrative laws outlining the AECs’ authority. In Sweden, a person found guilty of disregarding the rules of a certain task when acting on behalf of an authority may be subject to fines or up to two years in prison for misconduct, be it through intent or negligence, unless the misconduct is judged as slight [90] (Chapter 20: Section 1). However, how such possible mistakes by the AECs would come to result in any form of legal consequence for the committees or their chairpersons is unclear, as it is currently the role of the NAEC to review all projects categorized as “severe” and any other that the AECs may have demanded a review of. That same NAEC has admitted to not knowing how the “upper limit” should be applied.

A related issue concerns the reporting of national animal research statistics to the EU—namely, that possible transgressions of the “upper limit” of suffering are lost in the system. If the “severe” classification is found to have been exceeded, the responsible researchers are to report this “normally like any other use” under the “severe” category, and the reasons as to why the “severe” classification was exceeded should be included as commentary [51] (Annex II p. 7). However, this is where the system falters. The documents used by researchers in Sweden to report animal use, as well as the accompanying instructions, do not contain any mention of how to report cases where animals have been subjected to harm or suffering above the “upper limit” [91,92]. When contacted by the authors of this article, the Swedish 3Rs Center confirmed that the researchers are not informed or reminded of the “upper limit” and how transgressions of it are to be reported, but that they are able to add such information as comments should they wish (and know) to do so (Cecilia Bornestaf and Per Ljung, the Swedish 3Rs Center, personal communication, 14 April 2020). However, it may not always lie in the researcher’s interest to admit that a project for which they have been responsible has resulted in severe and unjustifiable suffering. Therefore, we believe it to be problematic that information requested for statistical purposes it is less nuanced than what would be needed to reflect compliance with legal requirements, risking the transparency of the ethical review process as well as the public’s trust in the system. The possibility, however small, that research animals may be subjected to severe prolonged suffering and that this is invisible in publically accessible statistics provides a false sense of comfort that animal research is better regulated and in some cases less harmful than is really the case. Additionally, this system of statistical gathering prohibits the SBA, via the NAEC, from knowing to the full extent how projects have turned out regarding degree of severity. We suggest a careful review of approved research projects given the “severe” classification by AECs but having transgressed the “upper limit”. This would guide the SBA as to where efforts should be directed to improve the ethical review and the severity assessment guidelines in particular. Are there patterns to which species’ suffering is most often underestimated? What kind of research has transgressed the limit? Are there certain AECs or researchers that seemingly lack the ability to correctly anticipate animal suffering? Seeing as there is no separate categorization or assembly of these projects today, this possibly beneficial information is lost.

### 4.4. Discrepancies between Law and Application Form/Insufficient Information

As our study has shown, the demands set by the Directive and their transcription into the L150 guidelines differ in several regards, as does the application form compared to the two aforementioned documents. As a result, what the Directive demands of the applicant could not, even if the application form was filled out to perfection, be satisfactorily provided by the applying researcher. In other words, the information requested by the application form and thus provided for the project evaluation is incomplete by nature and the AECs do thereby not receive enough information on which to base their decisions. As neither an HBA nor an overview of the 3Rs can be accurately performed without sufficient information [19,21], this risks the credibility of the Swedish ethical review process, public trust in the system, and the welfare of animals used in research when the Directive is not implemented and followed as is mandatory by all member states.

### 4.5. Public Interest and Transparency

As we discovered in our study, the AECs did not once include any mention in their decisions of having discussed the 3Rs. As such, even though one may hope and assume that the 3Rs were discussed, there is no evidence to support this. Neither did the AECs give satisfactory accounts of having conducted HBAs in more than four out of 18 cases. This is troubling. Similarly, the template “AECs Decision—Non-technical project summaries” is provided by the SBA for use by the AECs but was only included in four out of the 18 analyzed decisions. Using the template is not mandatory, but its inclusion would promote both constancy and transparency and in turn public insight. The reason(s) behind why the template was not included in more or all of the decisions has not been determined in our study. It is possible that the template is in fact used at committee meetings but for some reason is not incorporated amongst the final documents or that the AECs choose not to include it, nor any other document of corresponding content, unless the determined degree of severity differs from the one proposed by the researcher. Regardless, it is problematic from a transparency viewpoint that this step of the ethical review is not visible. Since applications submitted to, as well as decisions made by, AECs in Sweden are public documents, it could be anticipated that the lack of description of the 3Rs and HBA by the AECs may negatively impact public trust in animal-based research.

Seeing as the AECs are a part of, and therefore act in the name of, the Swedish Board of Agriculture, their decisions require motivating as specified by the Administrative Procedure Act [59] (Section 1, 20), [58] (Section 1, 32), Anders Elmström, Department of Animal Research at SBA, personal communication, 20 August 2020). What is less clear is how such a motivation should be constructed and how in depth it need be. This current uncertainty and allowance for interpretation as to how AEC decisions are to be motivated risks the integrity and transparency of the ethical review process, as well as the public’s insight and thereby trust in the system.

## 5. Conclusions

As is evident from our study, there are several areas within the ethical review of animal research in Sweden that are not working as intended.

Despite the limited number of applications included in this study, the revelation that not one single AEC included any mention of the 3Rs in their documented decisions is cause for great concern. So is the fact that the majority of the decisions did not contain a documented HBA. This is, we believe, largely due to the structure of the application form. By revising the application form (as further described below), the AECs would receive sufficient information to be able both to control that the 3Rs have been fulfilled by the applicants and to conduct a proper HBA. Furthermore, a clear expectation of a description of the 3Rs within the “technical body” of the application form would increase the awareness of the relevance of Replace, Reduce, and Refine for a proper HBA among both applicants and members of the ACEs. We further see a need of a revised common mandatory template for the AECs decisions where the inclusion of the 3Rs and a full HBA are clearly requested. This would not only facilitate the work of the AECs but also ensure increased transparency and public trust.

Furthermore, given the inadequate information provided by many researchers regarding the 3Rs and the description of potential or known harm due to housing and/or research procedures, we strongly recommend a review of the current education of researchers in Sweden. Judging by our results, it would be especially important to increase focus on the 3Rs, HBA, the “upper limit”, severe suffering and general awareness of precisely what information is to be provided in an application. This could be done through improving and strengthening the current mandatory laboratory animal science (LAS) courses and their subsequent continuing professional development (CPD) courses, and by ensuring competence is regularly updated, as required by the Directive [5] (Article 23, p. 2). To be concrete, all employees with function A and B eligibility could be required to demonstrate their competence and skills every third year so as to ensure good animal welfare, high quality research, and compliance with the Directive’s strive to continuously improve and streamline animal research procedures in line with the 3Rs. Additionally, research organizations and the scientific community as a whole could benefit greatly from adopting new approaches to increase and inspire 3R awareness and application among researchers, for example through revised organization and management strategies [77,93,94]. As it has been reported that researchers may prefer to consult other colleagues to increase their 3R knowledge rather than literature or databases [72], focusing on increasing the competence within the research community is highly recommendable. In addition, in line with the EU Directive, 3R support from animal welfare bodies may further strengthen researchers’ awareness and willingness to apply the 3Rs, although this is not yet fully implemented [95].

However, it is not only the researchers who would benefit from more and improved education judging by our results. AEC members evidently need more information on how to fulfill legal and ethical criteria through review of the 3Rs and performance of HBA. In addition to this, a better understanding among committee members of how to assess (and Refine) severe cases of suffering as well as how to apply the “upper limit” would ensure better protection for research animals, increased public trust and research quality, as well as, we believe, greater job satisfaction amongst AEC members. We also suggest that the AECs’ competence could be strengthened by adding expertise to knowledge and implementation of the different Rs through the employment or consultation of for example statisticians, system modelers, engineers, etc., and that the roles and impact of animal welfare bodies are strengthened. We further believe, similar to Hansen et al. [47], that the work of the AECs to ensure the application of the 3Rs could be improved and facilitated further through the inclusion of researchers in the committees who work with non-animal research and/or alternative methods.

Our study has also shed light on the difficulty of interpreting the “upper limit” of pain, suffering, or distress specified by the Directive. As such, we believe that it is urgent for all member states to create and adopt common guidelines and/or assessment templates specifically tailored to assess severe suffering and ensure that exceeding the “upper limit” is avoided at all costs. In addition, we believe it would be beneficial for animal welfare as well as the quality and transparency of animal research to place higher demands on researchers’ abilities to guarantee that suffering will not exceed the “upper limit” in studies that are categorized as severe. One possibly helpful way of regulating this could be to include in the application form a question of how long lasting the proposed suffering is expected to be. This is today already common practice in, for example, Germany where the categories “<1 day”, “1–7 days”, “7–30 days”, and “>30 days” are used [33]. Ideally, this would be accompanied by a consensus amongst member states concerning how to define “long-lasting” suffering and how the duration of suffering, pain, or distress should influence the severity assessment of a procedure.

In line with Alzmann [22], we suggest revisiting the idea of creating a comprehensive catalogue for harms (psychological and physical, present, past, and future) to be used when implementing the 3R and performing an HBA. Such a catalogue should preferably include but not be limited to, pain, suffering and distress due to the following factors: handling; genetic modification; social isolation or group size/dynamics; unnatural diurnal rhythms; room temperature, noise levels and other environmental factors; abrupt environmental changes; lack of environmental stimulation and/or inability to perform natural behavior and fulfill behavioral needs; anticipatory fear; and negative emotional states such as boredom, stress, and depression. For transparency and uniformity, the catalogue should be one and the same for all member states. Likewise, constructing common guidelines for all member states on how to better assess benefits would greatly improve and facilitate the work and task of the AECs, in turn ensuring trustworthy HBAs. However, achieving this, if at all possible, would warrant additional research.

As seen, there exist discernible discrepancies between the legal requirements of Directive 2010/63/EU, its implementation in Swedish legislation (SJVFS 2012:26), and the application form for project evaluation. As such, applying researchers are unable to provide the legally requested information concerning their proposed studies, and the AECs are forced to work with fragmented facts. In an effort to resolve this, a thorough comparison of the Directive and the L150 together with an exhaustive evaluation of which legal demands are and are not met by the current application form format will be conducted by the research group during 2021. This upcoming project will provide suggestions for how the application form may be revised to ensure that it corresponds to legal demands and is clear and easy to both fill out and read.

Additionally, due to the risk of violations of the “upper limit” getting lost in the statistics, we further believe that the system for reporting of actual degree of severity needs to be revised. One resolution could be to adopt an additional category of severity, higher than severe, solely for statistical use, so that any and all research whereby the “upper limit” has been exceeded is clearly visible, and strategies for addressing the causes for this can be applied. However, as long as it is the researchers themselves who are responsible for reporting the actual level of severity, we realize it may be difficult to motivate the use of such a category, especially if there would be reprimands involved due to the cause of unnecessary suffering and thereby the possible violation of both animal research and welfare legislation. This brings us to our next suggestion; in order to avoid bias and ensure transparency, we would like to suggest the implementation of an independent system for review of how the AECs have made and motivated their decisions, how they fulfilled the required points of analysis (3R, HBA, degree of severity, etc.), and how well the estimated degrees of severity correspond to the observed pain, distress, and suffering. This would motivate researchers to thoroughly plan for, implement, and demonstrate their use of the 3Rs (Refinement in particular). It would also stimulate the AECs to carry out a careful review of how the 3Rs have been applied and to perform in-depth HBAs. We would also like to open up the possibility for parties other than the researchers to contest decisions made by the AECs so as to ensure transparency, public trust, and that the welfare and intrinsic value of animals is safeguarded, valued, and respected.

It is today unclear exactly what is required for transparency to be fulfilled according to the Directive and to what extent the AECs’ decisions need to be motivated to live up to Swedish administrative law. What is clear is that the vast majority of the analyzed decisions of our study lacked transparency, as they did not include an account of how the 3Rs had been reviewed nor of a proper HBA. Thus, there is a need for in-depth interdisciplinary research into how the ethical aspects of the review process are to be applied and expressed so as to fulfill the transparency demands of the Directive and administrative legislation. Based on the results of such research, guidelines for motivating decisions could be formulated.

## Data Availability

Data publically accessible and available upon request from the Swedish district courts. Contact information available at: https://jordbruksverket.se/djur/ovriga-djur/forsoksdjur-och-djurforsok/forsoksdjur (5 March 2021).

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
