# Peer review of "Reviewing the Review: A Pilot Study of the Ethical Review Process of Animal Research in Sweden"

_animals, 2021, doi:10.3390/ani11030708_

Round 1

Reviewer 1 Report

pleas find my comments in the document attached

Reviewer 2 Report

Excellent paper. I only caught some typos that ought to be corrected:

347: delete "Third bullet" 

428: "judged as I for this criteria": add "inadequate" in parentheses as this is the first occurrence 

454: "Seven" shouldn't be capitalized 

464" "Neither that applications failing to account for one R did so to a greater extent for the other Rs" This sentence is missing some words though I'm not sure how the authors want it fixed.

1095: "given the weak performance of many researches" should be "researchers" 

1118: "to fulfil legal and ethical criteria through review of 3R and performance of HBA" should be "fulfill" and "3Rs" 

1125: "impact of animal welfare organs are strengthened" should be "organizations" 
